# Implicit Regularisation in Overparametrized Networks: A Multiscale Analysis of the Fokker-Planck equation

## Abstract

In over-parametrised networks, a large continuous set of solutions (an *invariant manifold*) exists where the empirical loss is minimal. However, noise in the learning dynamics can introduce a drift along this manifold, biasing the dynamics towards solutions with higher "smoothness", therefore acting as a regularizer. In Li et al. (2022), a derivation of this drift was presented, borrowing the results from Katzenberger (1991), which shows that in the small learning-rate limit, $\eta \to 0$, the learning dynamics can be approximated by a stochastic differential equation (SDE), whose solution exhibit two distinct phases: a first phase, occurring over a number of steps $O(\eta^{-1})$, where the parameters are deterministically driven towards the invariant manifold; and a second phase, over timescales $O(\eta^{-2})$, in which noise induces a deterministic drift along the invariant manifold. This latter contribution to the drift can be regarded as the result of averaging the dynamics over the $O(\eta^{1/2})$ fluctuations orthogonal to the manifold, described by an Ornstein–Uhlenbeck process, as first suggested by Blanc et al. (2020). We offer a new derivation of the results by Li et al. (2022), that builds on the very intuitive arguments of Blanc et al. (2020), by implementing the averaging of the Fokker-Planck equation associated with the $\eta \to 0$ dynamics over such Ornstein–Uhlenbeck quasi-stationary state. Our contribution demonstrates the application of multiscale methods for elliptic partial differential equations (PDEs) (Pavliotis & Stuart, 2008) to optimization problems in machine learning.

## 1 Introduction

For a model with parameters $\theta$, we consider stochastic update rules of the type

$$\hat{\theta}_{n+1} \leftarrow \hat{\theta}_n + \eta \Big( -\nabla_\theta L(\hat{\theta}_n) + \hat{\sigma}_{\hat{b}_n}(\hat{\theta}_n) \Big) \tag{1}$$

where $\hat{\sigma}_{\hat{b}}$ is a random error in the evaluation of the gradient of the average empirical loss $L$, due to the random choice of mini-batch $\hat{b}$, label noise, or the noise in the model, such as *Dropout* (Srivastava et al. (2014); see Appendix A.1 for details and notation).

In the general framework we are interested in, we do not distinguish between various sources of noise, and for our purposes it suffices to say that the learning dynamics can be written in the form of equation 1.

In the time coordinate $t = \eta^2 n$, which describes the learning dynamics over a number of steps $O(\eta^{-2})$, the increment is approximated by a stochastic differential equation (SDE) of the Langevin type, with a large drift $O(\eta^{-1})$:

$$\mathrm{d}\hat{\theta}_t = -\eta^{-1} \left. \nabla_\theta L(\theta) \right|_{\hat{\theta}_t} \mathrm{d}t + \sigma(\hat{\theta}_t) \cdot \mathrm{d}\hat{W}_t . \tag{2}$$

where $\mathrm{d}\hat{W}_t$ is a standard Wiener process, with multiplicative noise interpreted according to the Itô convention, and where the covariance of the noise (or diffusion tensor) is the symmetric matrix

$$\Sigma(\theta) = \sigma(\theta) \, \sigma(\theta)^\top \tag{3}$$

(see Appendix A.2 for details).

In overparametrized networks, there exist generally a manifold $\Gamma$ of points $\theta^*$, where the loss is minimum.

However, noise in the dynamics can induce a drift term along the manifold, which is often dubbed *implicit regularization*. In other words, noise in the learning dynamics can make the parameters $\theta$ drift towards regions of parameter space where the model is smoother.

Blanc et al. (2020) have first proposed that this drift is driven by an Ornstein-Uhlenbeck process taking place orthogonally to the manifold $\Gamma$, offering a local analysis that has a validity over $O(\eta^{1.6})$ steps.

Li et al. (2022), based on the main theorem from Katzenberger (1991), offer instead a global analysis, capturing the dynamics over $O(\eta^{-2})$ steps – that is through equation 2.

Here, we offer an alternative derivation based on the application of multiscale methods (*averaging*) to the Fokker-Planck equation associated with equation 2, which is in agreement and has the same validity as the results of Li et al. (2022).

## 2 MAIN RESULT

In this manuscript, we derive the effective stochastic dynamics at $O(\eta^{-2})$ steps by applying multiscale methods at the level of the *generator* of the dynamics in equation 2, i.e. through averaging of the associated Fokker-Planck equation. In this Section, we give an overview of the derivation, the details of which can be found in Appendix E.

The generator of the process $\hat{\theta}_t$ in equation 2 is defined as the partial differential operator

$$\mathcal{L} = -\frac{1}{\eta} \nabla_\theta L(\theta) \cdot \nabla_\theta + \frac{1}{2} \Sigma(\theta) : \nabla_\theta \nabla_\theta \ , \tag{4}$$

where $\Sigma = \sigma\sigma^\top$. If $\rho(\theta, t)$ is the probability density of the process at time $t$, the generator expresses the dynamics by describing rates of change of the probability density

$$\partial_t \rho = \mathcal{L}^* \rho \ , \tag{5}$$

where $\mathcal{L}^*$ is the adjoint of $\mathcal{L}$. This equation is the so-called Fokker-Planck equation (see Appendix C for details).

After an initial fast transient, due to the large drift $O(\eta^{-1})$ that quickly maps the the process from $\theta_0$ to $\theta^* = \Phi(\theta_0) \in \Gamma$, the process is bound to fluctuate in the vicinity of $\Gamma$, with orthogonal displacements $O(\eta^{1/2})$ – it can be verified that this is the correct scaling that ensures the balance between orthogonal drift and noise. Since $\nabla_\theta L = 0$ on $\Gamma$, the leading order is given by the linear (elastic) term in the perpendicular displacement from the manifold, $\delta\theta^\perp$, that is

$$\nabla_\theta L\big|_{\theta^* + \delta\theta^\perp} \simeq -H \cdot \delta\theta^\perp \ ,$$

where $H$ is the Hessian of the loss, evaluated at $\theta^*$.

We can introduce auxiliary variables $y = \eta^{-1/2}\delta\theta^\perp$, and expand $\mathcal{L}$ in the small parameter $\eta$. The leading order is given by $\eta^{-1}\mathcal{M}$, where $\mathcal{M}$ is the generator that describes the fast equilibration of the orthogonal fluctuations via the Ornstein-Uhlenbeck process,

$$\mathcal{M} = \left( -y^j H_j^k + \frac{1}{2} \Sigma_\perp^{kl} \frac{\partial}{\partial y^l} \right) \frac{\partial}{\partial y^k} \tag{6}$$

where $H$ is the Hessian of the loss $L$, $\Sigma_\perp = (1 - P)\Sigma(1 - P)$ (all terms evaluated at $\theta^* \in \Gamma$) and $P$ is the projector to the tangent space $T_{\theta^*}(\Gamma)$.

This process admits a Gaussian equilibrium steady state, given by

$$\mu_{\text{inv}}(y|\theta^*) = \frac{\det \Omega^{-1/2}}{(2\pi)^{M/2}} \exp\left\{ -\frac{1}{2} \Omega_{ij}^{-1} y^i y^j \right\} \tag{7}$$

where $\Omega$ is the solution of (Gardiner, 2009)

$$H(\theta^*)\,\Omega + \Omega\,H(\theta^*) = \Sigma_\perp(\theta^*)\,. \tag{8}$$

Here, the point on the manifold $\theta^*$ enters parametrically, as it is affected by the dynamics only at longer timescales. The coordinates of this point on $\Gamma$ therefore constitute the slow components of the system.

At the next non-vanishing order, $O(1)$, we find the generator

$$\mathcal{L}_0 = \Big( - y^j y^l \nabla H^i_{jl} + \frac{1}{2}\,\Sigma^{ij}\,P^l_j\,\frac{\partial}{\partial\theta^{*l}} \Big) P^k_i\,\frac{\partial}{\partial\theta^{*k}} \tag{9}$$

which describes the slow dynamics of $\theta^*$ along the manifold $\Gamma$. Note that the derivatives $\partial/\partial\theta^*$, projected onto the tangent space $T_{\theta^*}(\Gamma)$, can be expressed in terms of the derivatives with respect to the local coordinates $Z$.

We observe that the generator $\mathcal{L}_0$ depends on the fast variables $y$, which drive the first term . These, however, appear equilibrated at times $O(1)$, and can be averaged over the steady state $\mu_{\mathrm{inv}}(y|\theta^*)$. This simply amounts to replacing

$$\mathbb{E}_{y\sim\mu_{\mathrm{inv}}(\cdot|\theta^*)}[y^j y^l] = \Omega^{jl}$$

in equation 9 and obtain

$$\bar{\mathcal{L}}_0 = \Big( - P^k_i\,\Omega^{jl}\nabla H^i_{jl} + \frac{1}{2}\,\Sigma^{ij}\,P^l_j\,\frac{\partial P^k_i}{\partial\theta^{*l}} \Big)\frac{\partial}{\partial\theta^{*k}} + \frac{1}{2}\,\Sigma^{ij}\,P^l_j P^k_i\,\frac{\partial}{\partial\theta^{*l}}\frac{\partial}{\partial\theta^{*k}} \tag{10}$$

where we also explicitly expanded the last term with the second derivative, to highlight the dependence of $P$ on $\theta^*$.

From this average generator, one can read the effective dynamics

$$\mathrm{d}\hat\theta_t = \bar f(\hat\theta_t)\,\mathrm{d}t + \bar\sigma(\hat\theta_t)\cdot\mathrm{d}\hat W_t\,, \tag{11}$$

which lies on the manifold and where

$$\bar\sigma = P\sigma \tag{12}$$

$$\bar f = -P\,\nabla H : \Omega + \frac{1}{2}\Sigma : (P\cdot\nabla P) \tag{13}$$

or in components

$$\bar\sigma^i_k = P^i_j\,\sigma^j_k \tag{14a}$$

$$\bar f^i = -P^k_i\,\Omega^{jl}\nabla H^i_{jl} + \frac{1}{2}\,\Sigma^{ij}\,P^l_j\,\frac{\partial P^k_i}{\partial\theta^{*l}} \tag{14b}$$

with sum over the repeated indices.

Note that we assume that at every point on the manifold, the null-space of the Hessian $H$ is identified with the tangent space; that is, all the eigenvalues of $H$ with corresponding eigenvector orthogonal to $H$ are strictly positive, while all other eigenvalues are zero. In this assumption, the projector along the tangent space is related to $H$ via

$$HH^\dagger = H^\dagger H = 1 - P\,,$$

which allows to compute all terms explicitly in terms of the Hessian.

## 3 RELATED WORK

Equations 14 are consistent with the main result from Li et al. (2022).

The starting point for the analysis in Li et al. (2022) is the theorem by Katzenberger (1991) which states that over timescales $O(1)$, equation 2 can be approximated by

$$\mathrm{d}\hat\theta^i_t \simeq \mathrm{d}\Phi^i(\hat\theta) = (\partial_j\Phi^i)\,\sigma^j_k\cdot\mathrm{d}\hat W^k_t + \frac{1}{2}(\partial_j\partial_k\Phi^i)\,\Sigma^{jk}\,\mathrm{d}t \tag{15}$$

where $\Phi$ is the map that yields the stationary solution of the deterministic part of the dynamics, as a function of the initial condition. Here, and in the following, the Einstein convention of the summation over repeated indices is used. Equation 15 expresses the intuition that, when there is a large drift (as in equation 2), whenever $\theta$ deviates slightly from the invariant manifold, it is quickly "zapped" onto the manifold, at $\Phi(\theta)$, and a slow motion along the manifold can be described by applying Itô formula through the map $\Phi$. Li et al. (2022) apply this result to the gradient flow dynamics where the loss $L$ has a manifold $\Gamma$ of minimizers, and explicitly provide all terms in equation 15 in terms of the gradient noise covariance $\Sigma$ and the higher order (second and third) derivatives of the loss $L$.

In particular, they show that the Jacobian of the map $\Phi$, evaluated at a point $\theta^*$ on the invariant manifold is the projector along the tangent space to the manifold at that point. Therefore, the noise term in equation 15 is simply the component of the noise along the tangent space.

The second derivatives of $\Phi$, contracted with the noise covariance $\Sigma$, give the deterministic part of the noise-induced drift:

$$\left(\partial_j \partial_k \Phi^i\right) \Sigma^{jk} = -(H^\dagger)^i_j \, (\nabla H)^j_{kl}(\Sigma_\parallel)^{kl} - P^i_j \, (\nabla H)^j_{kl} \left(2 \, (H^\dagger)^k_n (\Sigma_{\perp,\parallel})^{nl} + \Omega^{kl}\right) \tag{16}$$

where $H^\dagger$ is the pseudo-inverse of the Hessian of the loss, $H$; $\nabla H$ is the gradient of the Hessian; $P$ is the projector on the tangent space; $\Sigma_\parallel = P\Sigma P$, $\Sigma_{\perp,\parallel} = (1-P)\Sigma P$ and $\Omega$ is the solution of the Lyapounov equation

$$H \, \Omega + \Omega \, H = \Sigma_\perp \tag{17}$$

with $\Sigma_\perp = (1-P)\Sigma(1-P)$.

## 4    DISCUSSION

In this short contribution, we offer an alternative derivation of the results presented in Li et al. (2022), that uses the averaging of the Fokker–Planck equation associated with the learning dynamics in the limit $\eta \to 0$. This method rests on the solid mathematical foundation of multiple-scale methods for elliptic partial differential equations (Pavliotis & Stuart, 2008). The result of this analysis is in agreement with the those presented in Li et al. (2022), borrowing the theorem from Katzenberger (1991) which provides a proof relying on Itô calculus.

Multiscale methods find numerous applications in various fields in the natural sciences (E, 2011; Bo & Celani, 2017). In this manuscript, we have shown an application that is of interest for the machine learning community – i.e. as a tool to derive the approximate dynamics at long times, when a manifold of optimizers $\Gamma$ is present, featuring a deterministic term that plays the role of an implicit regularization.

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

# A  BACKGROUND

In this Appendix, we lay out the main definitions and notation needed throughout the manuscript.

We will focus on a standard statistical learning task. For the sake of generality, and as we are interested in the noise during learning, we will define a notation that encompasses all possible types of noise in this setting.

## A.1  STOCHASTIC LEARNING DYNAMICS

Given a dataset $\mathcal{D}$ with $p$ input-output pairs, $\mathcal{D} = \{X^\mu, y^\mu\}_{\mu=1}^p$, where $X^\mu$ indicates the input, and $y^\mu$ the output (or label) for the pair $\mu$.

In the case of *label noise*, a 0-mean random variable [1] $\hat{\varepsilon}$ is added to the output

$$\hat{y}^\mu = y^\mu + \hat{\varepsilon} \,, \tag{A1}$$

We are given a model parametrised by $\theta \in \mathbb{R}^D$, and perturbed by some noise $\hat{\xi}$:

$$\hat{f}^\mu(\theta) = f(X^\mu; \theta, \hat{\xi}) \,. \tag{A2}$$

For instance, $f$ if often represented by an artificial neural network (ANN), and the noise in *Dropout* (Srivastava et al., 2014), $\hat{\xi} \equiv \{\hat{\xi}_i\}$, a collection of i.i.d. Bernoulli random variables multiplying the activation of each hidden unit in the network, labelled by $i$; in *DropConnect* (Wei et al., 2020), $\hat{\xi} \equiv \{\hat{\xi}_{ij}\}$, that is i.i.d. Bernoulli random variables multiplying each weight in the network.

One can cast the statistical learning task into the minimization of the loss (or empirical risk)

$$\hat{L}(\theta, \mathcal{D}) = \frac{1}{p} \sum_\mu^p \hat{\ell}^\mu(\theta) \,, \tag{A3}$$

i.e. the average over the dataset $\mathcal{D}$ of the single-data point loss

$$\hat{\ell}^\mu(\theta) = \ell(\hat{f}^\mu(\theta), \hat{y}^\mu) \,, \tag{A4}$$

with respect to the parameter vector $\theta$.

However, due to the presence of noise, the optimization problem as such is not well-defined. We can choose to minimize the *mean* of $\hat{L}$ over the various sources of noise:

$$L(\theta, \mathcal{D}) = \mathbb{E}_{\hat{\xi}, \hat{\varepsilon}}\big[\hat{L}(\theta, \mathcal{D})\big] \,. \tag{A5}$$

One algorithmic way to seek for the minimum of the loss equation A5 is via gradient descent (GD). That is, by choosing a small-enough learning rate $\eta$, [2]

$$\hat{\theta}_{n+1} \leftarrow \hat{\theta}_n - \eta \, \nabla_\theta L(\theta)\big|_{\hat{\theta}_n} \tag{A6}$$

with $n$ the number of updates since the start of the learning.

However, in practice, it is common to use a stochastic update rule,

$$\hat{\theta}_{n+1} \leftarrow \hat{\theta}_n - \frac{\eta}{|\hat{b}_n|} \sum_{\nu \in \hat{b}_n} \nabla_\theta \hat{\ell}^\nu(\theta) \equiv \hat{\theta}_n - \eta\Big(g(\hat{\theta}_n) + \hat{\sigma}_{\hat{b}_n}(\hat{\theta}_n)\Big) \tag{A7}$$

where $\hat{\sigma}_b$ is a noise term, accounting for the deviation of the batch estimator of the gradient and the true gradient.

---

[1] All random variables, are indicated with a "hat" symbol, e.g. $\hat{X}$. Their value, or realisation, is without the "hat".

[2] Hereafter, we drop the dependence on $\mathcal{D}$ of the empirical loss.

## A.2 LANGEVIN DYNAMICS APPROXIMATION

By iterating equation 1 over $\Delta n$ steps, the change in the weights is

$$\hat{\theta}_{n+\Delta n} - \hat{\theta}_n = \eta \sum_{m=1}^{\Delta n} \left( g(\hat{\theta}_{n+m}) + \sigma_{\hat{b}_{n+m}}(\hat{\theta}_{n+m}) \right) \tag{A8}$$

If $\eta \ll 1$, so that that after $\Delta n$ steps weights change so little that we can neglect changes in $g$ and in the statistics of $\hat{\sigma}_b$, then we can approximate

$$\hat{\theta}_{n+\Delta n} - \hat{\theta}_n \simeq \eta \, \Delta n \, g(\hat{\theta}_n) + \eta \, \sigma(\hat{\theta}_n) \sum_{m=1}^{\Delta n} \hat{\epsilon}_{n+m}$$

where $\hat{\epsilon}_n$ is uncorrelated, standardised noise of the gradient, i.e. $\mathbb{E}_{\hat{\xi},\hat{\varepsilon},\hat{b}}\left[\hat{\epsilon}_n\right] = 0$, $\mathbb{E}_{\hat{\xi},\hat{\varepsilon},\hat{b}}\left[\hat{\epsilon}_n\hat{\epsilon}_m^\top\right] = \delta_{nm}\,\mathbb{1}$, so that the covariance of the noise in the gradient estimation is

$$\mathbb{E}_{\hat{\xi},\hat{\varepsilon},\hat{b}}\left[\hat{\sigma}_{\hat{b}}(\theta)\,\hat{\sigma}_{\hat{b}}(\theta)^\top\right] = \sigma(\theta)\,\sigma(\theta)^\top \equiv \Sigma(\theta) \,. \tag{A9}$$

If we assume further that $\eta \ll 1$ and $\Delta n \gg 1$, but in such a way that $\eta\Delta n \ll 1$, then, by the central limit theorem the noise term converges to Gaussian random variable with 0 mean and covariance $\Delta n\,\Sigma(\hat{\theta}_n)$:

$$\hat{\theta}_{n+\Delta n} - \hat{\theta}_n \simeq \eta \, \Delta n \, g(\hat{\theta}_n) + \eta \, \sqrt{\Delta n} \, \sigma(\hat{\theta}_t) \, \hat{\omega}_n$$

where $\hat{\omega}_t$ is a standard Gaussian random variable.

We can re-scale the time coordinate by introducing the continuous time variable $t = \eta\,\Delta n$, so that the learning dynamics can be approximated by a stochastic differential equation (SDE), where in the infinitesimal time step $dt = \eta\,\Delta n$ the weights undergo the increment

$$d\hat{\theta}_t = g(\hat{\theta}_t)\,dt + \eta^{1/2}\,\sigma(\hat{\theta}_t)\cdot d\hat{W}_t \tag{A10}$$

where $\hat{W}_t$ is the Wiener process, i.e. $\mathbb{E}\left[d\hat{W}_t\right] = 0$ and $\mathbb{E}\left[d\hat{W}_t\,d\hat{W}_{t'}\right] = dt\,\delta(t-t')\mathbb{1}$, and where the multiplication symbol "$\cdot$" is used to indicate the Itô convention for the integration of the noise. [3] Equation A10 describes the limiting dynamics over $O(\eta^{-1})$ steps. All that is required is the mean and the covariance of the stochastic increments $\hat{\Delta}\theta$.

We realise that the noise term is scaled by $\eta^{1/2}$, which is small (while both $g$ and $\sigma$ are $O(\eta^0)$).

Note that $d\hat{W}$ scales as $dt^{1/2}$ under re-scaling of the time coordinate, so that under the change of variable $\tilde{t} = \eta\,t$, we can re-write equation A10 as

$$d\hat{\theta}_{\tilde{t}} = -\eta^{-1}\,g(\hat{\theta}_{\tilde{t}})\,d\tilde{t} + \sigma(\hat{\theta}_{\tilde{t}})\cdot d\hat{W}_{\tilde{t}} \,. \tag{A11}$$

In time units capturing the dynamics over timescales corresponding to $O(\eta^{-2})$ discrete updates, then, the Langevin equation features a large drift term, $O(\eta^{-1})$, that "zaps" the parameters onto the invariant manifold.

## B INTUITION FOR THE MAIN RESULT

In the large-drift limit of the Langevin SDE, equation A11, the noise become non-negligible only when the drift is of the same order or smaller. This can occur close to the invariant manifold, which is characterised by vanishing drift.

This implies that there are 2 distinct phases in the learning dynamics: *first*, follow the deterministic dynamics until we are close to the invariant manifold (fast); *then* follow the full stochastic dynamics (slow).

---

[3] Hereafter, we use the notation "$\mathbb{E}$" to denote expectations over the Wiener process, which accounts for all sources of noise (label noise $\hat{\varepsilon}$, model noise $\hat{\xi}$ and SGD noise $\hat{b}$). Also, in this case, the stochastic dynamics is non-anticipative, and the noise has to be interpreted according to the Itô convention.

After the first (deterministic) phase, we reach a point $\theta^*$ on the invariant manifold, which depends on the initial conditions.

Then, since the noise is small compared to the drift, the dynamics will be confined to be close to the invariant manifold. This allows one to expand the drift term in equation A11 in the displacement $\delta\theta = \theta - \theta^*$:

$$\nabla L\big|_\theta \simeq H(\theta^*)\,\delta\theta + \frac{1}{2}\,\nabla H(\theta^*) : \delta\theta\,\delta\theta^\top \tag{B1a}$$

or in components

$$\frac{\partial L}{\partial \theta^i}\bigg|_\theta \simeq H(\theta^*)^i_j\,\delta\theta^j + \frac{1}{2}\,(\nabla H(\theta^*))^i_{jk}\,\delta\theta^j\,\delta\theta^k \tag{B1b}$$

where $H$ and $\nabla H$ denote the Hessian of the loss and its gradient, respectively, and where we used the fact that $\nabla L|_{\theta^*} = 0$. The Hessian of the loss, $H$, evaluated at a point on the manifold, $\theta^*$ must be non-negative, and its kernel is assumed to be the tangent space; this assumption means that there are no soft modes in the perpendicular fluctuations, i.e. 0 eigenvalues of $H$ with eigenvectors perpendicular to the manifold. Therefore, in the first order approximation, there is always a non-vanishing component of $g$ acting only orthogonally to the invariant manifold. Any contribution to the force along the manifold will enter only at the second order in the expansion in the displacement, and therefore we expect them to produce an effect at even longer time scales.

We can write the Langevin SDE A11 by replacing the expansion in equation B1, and project the equation along the directions tangent and orthogonal to the invariant manifold:

$$\mathrm{d}(\delta\hat\theta)^\| \equiv \mathrm{d}\theta^* = -\eta^{-1}\,(\nabla H : \delta\hat\theta\delta\hat\theta^\top)^\|\,\mathrm{d}t + (\sigma \cdot \mathrm{d}\hat W_t)^\| \tag{B2a}$$

$$\mathrm{d}(\delta\hat\theta)^\perp = -\eta^{-1}\,H(\delta\hat\theta)^\perp\,\mathrm{d}t + (\sigma \cdot \mathrm{d}\hat W_t)^\perp - \eta^{-1}\,(\nabla H : \delta\hat\theta\delta\hat\theta^\top)^\perp\,\mathrm{d}t \tag{B2b}$$

The first two terms in equation B2b constitute a multivariate *Ornstein-Uhlenbeck* process in the perpendicular direction, which equilibrates over timescales $O(\eta)$, yielding a Gaussian steady-state with 0 mean and fluctuations $O(\eta^{1/2})$. If one replaces these fluctuations in the drift term of equation B2a, one notices that it yields a term $O(1)$ parallel to the manifold (provided that the Hessian has non-vanishing derivative along the tangent space).

Li et al. (2022) prove that all the terms involving $\nabla H$ are $O(1)$ and provide explicit expressions for them; together with the tangent noise term $(\sigma \cdot \mathrm{d}\hat W_t)^\|$, these deterministic terms are non-negligible contributions to the dynamics over $O(\eta^{-2})$ steps in the learning dynamics.

## C    THE GENERATOR AND THE FOKKER-PLANCK EQUATION

Given a Markov process $\hat\theta_t$, we can denote $\rho_{1|1}(\theta', t'|\theta, t)$ the conditional probability density function for $\hat\theta_{t'} = \theta'$ given $\hat\theta_t = \theta$ (also called *propagator*). For an arbitrary $t'' \in (t, t')$, this satisfies

$$\rho_{1|1}(\theta', t'|\theta, t) = \int \mathrm{d}\theta''\,\rho_{1|1}(\theta', t'|\theta'', t'')\,\rho_{1|1}(\theta'', t''|\theta, t)\,, \tag{C1}$$

which expresses the conservation of probability for the Markov process.

For continuous-time processes, we may take $t'' \to t^+$, and cast equation C1 into a first-order differential equation in time, that is

$$\partial_t \rho_{1|1} + \mathcal{L}\rho_{1|1} = 0\,, \tag{C2}$$

where the operator $\mathcal{L}$ is called *generator* of the process. Equation C2 is the *backward* form of the *Kolmogorov* (or *master*) *equation*, and describes the dynamics through its dependence on the initial condition at time $t$, $\hat\theta_t$.

For diffusive processes, small time increments imply small increments in $\hat\theta_t$. In these cases, it is possible to take the limit of infinitesimal increments to cast equation C1 into a partial differential equation (PDE), where $\mathcal{L}$ is a differential operator on the state variables. For the dynamics described by the Langevin SDE, equation A11, if $\hat\theta_t = \theta$, then $\hat\theta_{t+\mathrm{d}t}$ is Gaussian-distributed with mean $\theta - g(\theta)\,\mathrm{d}t$ and covariance $\Sigma(\theta)\,\mathrm{d}t$ (see, e.g. Gardiner (2009)). In this case, the generator is

$$\mathcal{L} = -\frac{1}{\eta}\,g(\theta) \cdot \nabla_\theta + \frac{1}{2}\,\Sigma : \nabla_\theta\nabla_\theta\,. \tag{C3}$$

In equation C2 it operates on the $\theta$ variables, while $(\theta', t')$ enter parametrically, that is

$$\partial_t \rho_{1|1} - \frac{1}{\eta} g^i \frac{\partial}{\partial \theta^i} \rho_{1|1} + \frac{1}{2} \Sigma^{ij} \frac{\partial^2}{\partial \theta^i \, \partial \theta^j} \rho_{1|1} = 0 \tag{C4}$$

where $g$ and $\Sigma$ are evaluated at $\theta$. Equation C4 is known as the backward *Fokker-Planck equation* (FPE).

Analogously, one can take the limit $t'' \to t'^{-}$ in equation C1, and obtain the FPE in its *forward* form, i.e. by describing the dynamics via the dependence of $\rho_{1|1}$ as the measure over $\hat{\theta}_{t'}$:

$$\partial_{t'} \rho_{1|1} - \mathcal{L}^* \rho_{1|1} = 0 \tag{C5}$$

where $\mathcal{L}^*$ is the adjoint of $\mathcal{L}$, and it operates on the variables $\theta'$, while the dependence on $(\theta, t)$ is parametric. Similarly, by taking the limit of small increments in equation C5, one obtains the forward FPE,

$$\partial_{t'} \rho_{1|1} = \frac{1}{\eta} \frac{\partial}{\partial \theta'^i} (g^i \rho_{1|1}) + \frac{1}{2} \frac{\partial^2}{\partial \theta'^i \, \partial \theta'^j} (\Sigma^{ij} \rho_{1|1}) \tag{C6}$$

where $g$ and $\Sigma$ are evaluated at $\theta'$.

Due to the linearity of $\mathcal{L}$, it follows from equation C2 that given any function $F(\hat{\theta}_{t'})$, its expectation over $\rho_{1|1}$,

$$h(\theta, t) = \mathbb{E}_{\hat{\theta}_{t'}} \big[ F(\hat{\theta}_{t'}) \, \big| \, \hat{\theta}_t = \theta \big] = \mathbb{E}_{\theta' \sim \rho_{1|1}(\cdot, t' | \theta, t)} \big[ F(\theta') \big] \tag{C7}$$

also satisfies equation C2.

### NULL SPACE OF THE GENERATOR

We see from equation C5, that if an invariant measure $\mu_{\text{inv}}$ exists, it satisfies

$$\mathcal{L}^* \mu_{\text{inv}} = 0 \, , \tag{C8}$$

or, in algebraic terms, it is found as the left null vector of the generator $\mathcal{L}$. If the process is ergodic, a single stationary solution exists, and the invariant measure is unique. In this case, the corresponding right eigenvector $r$, has all equal entries, i.e. equals a constant everywhere. From the normalization condition of the invariant measure, this constant is 1, which ensures the biortonormality condition

$$\langle \mu_{\text{inv}}, r \rangle = \int \mathrm{d}\theta' \, \mu_{\text{inv}}(\theta') = 1 \, . \tag{C9}$$

If the process is not ergodic, then multiple invariant measures, $\{\mu_{\text{inv}}^a\}$ exist which depend on the initial conditions – corresponding to as many left null vectors of $\mathcal{L}$, $\{r^a\}$. This is the case where the dynamics has absorbing states, that is $\{\theta^a\}$ with vanishing exit probability rates. In such cases, then, the right null vector $r^a$ corresponding to $\mu_{\text{inv}}^a$ gives the probability that $\hat{\theta}_t$ is absorbed at $\theta^a$ (i.e. it reaches $\theta^a$ before any other absorbing state) as a function of the initial condition.

## D  OVERVIEW OF MULTISCALE METHODS: DECIMATION AND AVERAGING

In this Appendix, we provide an overview of multiscale methods for Markov processes. In particular, we introduce decimation and averaging for continuous-time Markov processes. Although in this Appendix we discuss the simpler case of discrete states, the treatment presented also applies to continuous-states case (diffusive systems), which is of interest for this manuscript and will be presented in Appendix E. For this reason, we keep the notation as general as possible.

We present a rather informal introduction. For a more detailed exposition with applications see Bo & Celani (2017) and Pavliotis & Stuart (2008).

### D.1  TIMESCALE-SEPARATED SYSTEMS: FAST AND SLOW VARIABLES

Let us consider a continuous-time Markov process $\hat{\theta}_t$ in which we can identify a slow component $\hat{Z}_t$ and a fast component $\hat{Y}_t$,

$$\hat{\theta}_t = (\hat{Z}_t, \hat{Y}_t) \, , \tag{D1}$$

The separation between fast and slow variables means that we can identify a small parameter $\epsilon \ll 1$ that quantifies the rates of change in $\hat{Z}_t$ relative to those in $\hat{Y}_t$.

In the discrete case, this means that states are grouped into "blocks", labelled by the slow variables $Z$. Transitions *across* blocks occur with a frequency $O(\epsilon^0)$. *Within* each block, states are further identified by the fast variables $Y$, among which transitions happen with high frequency, $O(\epsilon^{-1})$.

In the continuous case, the interpretation of $Z$ and $Y$ depends on the application. For the sake of the exposition, we can think of the simplest case where both fast and slow variables have a drift-diffusion dynamics, and where $\hat{\theta}_t \in \mathbb{R}^D$ can be written as the direct sum between and $\hat{Z}_t \in \mathbb{R}^{D-M}$ and $\hat{Y}_t \in \mathbb{R}^M$, as in equation D1. This extends to the case of Riemannian geometry, where $\hat{Z}_t$ are the coordinates of a $(D-M)$-dimensional manifold in $\mathbb{R}^D$.

Over a time-scale $O(\epsilon)$, therefore, the stochastic dynamics of $\hat{Y}_t$ can be regarded as *conditioned* on $\hat{Z}_t$. In certain cases, the fast dynamics involves states that are *transient*, and therefore visited only for a short time, $O(\epsilon)$. Then, these states are effectively "eliminated", and they do not feature in the dynamics at timescales $O(1)$. In the jargon of multiscale methods, this elimination of states takes the name of *decimation*.

Over longer timescales, $O(\epsilon^0)$, by the time a transition occurs to a different block, the fast variables can be assumed to have reached a steady state within the starting block. Then, the probability per unit time (or transition rate) between blocks can be approximated by the average of the transition rates over this steady state. In the multiscale jargon, the operation of computing the effective dynamics at timescales $O(1)$ as the average of the slow dynamics over the fast variables is referred to as *averaging*.

In diffusive systems, averaging can yield a trivial dynamics where both average drift and average diffusivity vanish. In this case, a non-trivial dynamics can be found at even longer timescales, $O(\epsilon^{-1})$, by applying a further procedure called *homogenization* – given by computing the solvability condition at the second order in $\epsilon$. We are not concerned with this situation in this manuscript; see Pavliotis & Stuart (2008) and Bo & Celani (2017) for more in-depth discussion and examples.

### D.2 MULTIPLE TIMESCALE EXPANSION

In this section, we translate the intuition developed in Appendix D.1 into algebra.

The time-scale separation described above, with relative frequency parameter $\epsilon \ll 1$, means that we can express the generator as

$$\mathcal{L} = \epsilon^{-1}\mathcal{M} + \mathcal{L}_0 \,, \tag{D2}$$

where both $\mathcal{M}$ and $\mathcal{L}_0$ contains terms that do not scale with $\epsilon$.

In the discrete case, $\mathcal{L}$ is a matrix containing the transition probabilities per unit time (rates). Fast and slow variables can be identified if the leading order term, $\mathcal{M}$, is a reducible matrix, i.e. it is possible to find a permutation of the state indices such that $\mathcal{M}$ acquires a block-triangular structure. In this case, one can introduce auxiliary variables $Z$ to label the blocks (as discussed above). Then, the first-order term, $\mathcal{L}_0$, contains transition rates outside of the diagonal blocks.

There can be blocks, however, which are not probability conserving, under the generator $\mathcal{M}$, i.e. contain transient states for the fast dynamics. In this case, the $Z$ variables associated to these blocks are redundant (as they do not label blocks that "survive" at times $O(1)$) and can be eliminated. This is what is referred to as *decimation* in the multiscale literature (Bo & Celani, 2017, Part I, Sec. 2).

Let us consider the problem

$$\partial_t h + \mathcal{L}h = 0 \,. \tag{D3}$$

Due to the expression of the generator $\mathcal{L}$ as an expansion in the small parameter $\epsilon$, we can seek for a perturbative form of the solution,

$$h^{(\epsilon)} = h^{(0)} + \epsilon\, h^{(1)} + \ldots \tag{D4}$$

where $h^{(0)}$ is the solution of the equation D3 where only the leading term is retained, and $h^{(1)}$ enters as a small perturbation. Correspondingly, we introduce an auxiliary time variable $\tau$ that captures the

dynamics at timescales $O(\epsilon)$, in addition to the natural time variable $t$ that we use to describe the dynamics at timescales $O(1)$:

$$t \to \epsilon\tau + t \, . \tag{D5}$$

This implies that the time derivative is expressed as

$$\partial_t \to \epsilon^{-1} \, \partial_\tau + \partial_t \, . \tag{D6}$$

By combining equations D3–D6, we obtain a hierarchy of equations to be solved iteratively:

$$O(\epsilon^{-1}): \, (\partial_\tau + \mathcal{M})h^{(0)} = 0 \tag{D7a}$$

$$O(1): \, \mathcal{M}h^{(1)} = -(\partial_t + \mathcal{L}_0)h^{(0)} \tag{D7b}$$

Note that the solution at $O(\epsilon^{-1})$ can be inserted as the known term in the equation at $O(1)$, which can be solved to obtain a closed form solution up to next-to-leading order in the expansion in $\epsilon$.

After a short transient, the leading-order equation, equation D7a, is

$$\mathcal{M}h^{(0)} = 0 \tag{D8}$$

The leading-order term in the perturbative solution, then, can be found as a right null vector of the fast part of the generator, $\mathcal{M}$. Note that this is guaranteed to exist, provided that the fast dynamics, conditioned on the slow variables, admits a steady state distribution $\mu_{\mathrm{inv}}$, which is a left null vector of $\mathcal{M}$ as discussed in Appendix C,

$$\mathcal{M}^* \mu_{\mathrm{inv}}(\cdot|Z) = 0 \, , \tag{D9}$$

where the notation $v(\cdot|Z)$ indicates a vector indexed by (or a function of) the state variables, parametrically dependent on $Z$. Here we see that the slow variables $Z$ label the steady states of the fast dynamics: this is because there is one "absorbing block" of states for each value of $Z$ which survives the decimation step. Correspondingly, there exists a right null vector of $\mathcal{M}$, for each non-decimated $Z$, which we denote $r(\cdot|Z)$, defined in such a way to satisfy the biortonormality condition

$$\langle \mu_{\mathrm{inv}}(\cdot|Z), r(\cdot|Z') \rangle = \delta_{Z,Z'} \tag{D10}$$

Then, the general solution of equation D8 is a linear combination

$$h^{(0)} = \sum_Z \psi(Z) \, r(\cdot|Z) \, . \tag{D11}$$

We can therefore proceed to solve the next-to-leading order, equation D7b, which formally involves the inverse of $\mathcal{M}$. However, since we know that $\mathcal{M}$ has a non-trivial kernel, we need to invoke a solvability condition, or *Fredholm alternative*, which consists in imposing that the known term in equation D7b lies in the space orthogonal to the kernel of $\mathcal{M}^*$.

$$\langle \mu_{\mathrm{inv}}, (\partial_t + \mathcal{L}_0)h^{(0)} \rangle = 0 \tag{D12}$$

This can be obtained by taking the inner product of equation D7b with $\mu_{\mathrm{inv}}(\cdot|Z)$ from the left, and using the fact that

$$\langle \mu_{\mathrm{inv}}, \mathcal{M}h^{(1)} \rangle = \langle h^{(1)}, \mathcal{M}^*\mu_{\mathrm{inv}} \rangle = 0 \, . \tag{D13}$$

By replacing the general solution D11 into equation D12, and by using the biorthonormality condition, equation D10, we have that for every $Z$

$$\left(\partial_t + \bar{\mathcal{L}}_0\right)\psi(Z) = 0 \tag{D14}$$

where

$$\bar{\mathcal{L}}_0 \equiv \bar{\mathcal{L}}_0(Z) = \langle \mu_{\mathrm{inv}}(\cdot|Z), \mathcal{L}_0 r(\cdot|Z) \rangle \tag{D15}$$

This is the algebraic expression of the decimation and averaging procedure described in the previous section. Equation D15 states that the operator $\bar{\mathcal{L}}_0$ is nothing but the generator containing the average of the transition rates between blocks over the conditional distribution $\mu_{\mathrm{inv}}(\cdot|Z)$. Then, the solvability condition for the next-to-leading order term is equivalent to the Kolmogorov equation for the slow variables, obtained as the average of the slow dynamics over the fast variables.

In Appendix E, we will discuss the application of this procedure to the large-drift Langevin SDE, describing the long-time stochastic learning dynamics in the small-learning rate limit.

# E  SINGULAR PERTURBATION THEORY OF THE FOKKER-PLANCK EQUATION

In this Appendix, we describe the averaging procedure for the FPE associated with the Langevin SDE, equation A11. We follow similar steps as for the general case in Appendix D, but highlighting some of the peculiarities of the analysis for a diffusive process with an invariant manifold $\Gamma$.

This will allow us to derive the main result presented in Section 2.

## E.1  EXPANSION IN MULTIPLE TIMESCALES

The Fokker-Planck generator $\mathcal{L}$, given in equation C3, can be expanded in powers of a small parameter $\epsilon$,

$$\mathcal{L} = \epsilon^{-1}\mathcal{M} + \mathcal{L}_0 \ . \tag{E1}$$

In equation C3 the small parameter is identified by $\epsilon = \eta$, the small learning rate, and

$$\mathcal{M} = -g \cdot \nabla_\theta \tag{E2}$$

$$\mathcal{L}_0 = \frac{1}{2}\,\Sigma : \nabla_\theta \nabla_\theta \tag{E3}$$

As mentioned in Appendix D.2, under certain conditions on the generator of the fast dynamics, $\mathcal{M}$, we can make the identification between fast variables $Y$ and slow variables $Z$, so that we can express $\theta \equiv (Z, Y)$.

### E.1.1  REDUCIBILITY OF THE FAST GENERATOR – FAST AND SLOW VARIABLES

In Appendix D.2, we highlighted that the condition for the identification of fast and slow variables is the reducibility of the generator $\mathcal{M}$. In the discrete-states case, this is the possibility to permute the state indices and reduce $\mathcal{M}$ into a block-triangular transition-rates matrix.

In this case, $\mathcal{M}$ is a differential operator, and the reducibility condition requires more care. However, we can recognise that $\mathcal{M}$ in equation E2 is the transport operator along the flow of the deterministic dynamic $\mathrm{d}\theta/\mathrm{d}\tau = -g(\theta)$. Given an initial condition $\theta_0$ at $\tau = 0$, we denote the solution of the dynamical system by the map

$$\phi(\theta_0, \tau) \ , \tag{E4}$$

i.e. a function of both $\tau$ and the initial condition. We assume that in the limit $\tau \to \infty$,

$$\theta^* = \Phi(\theta_0) = \lim_{\tau \to \infty} \phi(\theta_0, \tau) \tag{E5}$$

lies on the invariant (minimum loss) manifold $\Gamma$.

If the loss is not singular, different flow lines will not cross. Then, for each point $\theta^* \in \Gamma$, one can therefore identify a manifold $\Theta(\theta^*)$ containing all the initial conditions that lead to the same point $\theta^*$. By definition, the manifold $\Theta(\theta^*)$ is spanned by the flow lines terminating at $\theta^*$: If we parametrize $\Gamma$ with local coordinates $Z$ we can unequivocally denote this manifold $\Theta(Z)$.

Similarly, we can introduce local coordinates $Y$ that parametrize the manifolds $\Theta(Z)$, conveniently chosen so that $\Gamma$ is identified by $Y = 0$. Any point $\theta$ is then identified by $(Z, Y)$. [4]

Importantly, since the flow lines of the deterministic dynamics are locally tangent $g = \nabla_\theta L$, they are locally orthogonal to the level sets of $L$. As a consequence, they are locally orthogonal to the $\Gamma$ itself: more precisely, this means that at any point $\theta^* = (Z, 0) \in \Gamma$, the tangent space to $\Gamma$, $T_{\theta^*}(\Gamma)$, is the orthogonal complement of the tangent space to $\Theta(Z)$,

$$T_{\theta^*}(\Gamma) = T_{\theta^*}\big(\Theta(Z)\big)^\perp \ . \tag{E6}$$

In conclusion, we noticed that the fast dynamics does not affect the coordinates $Z$ of the invariant manifold $\Gamma$, while it drives the coordinates $Y$ of the manifold $\Theta(Z)$ to 0, i.e. $\theta$ to $\theta^* = (Z, 0) \in \Gamma$. The manifolds, labelled by $Z$, $\Theta(Z)$, play the role of the "blocks" of Appendix D, coupled by the

---

[4]More precisely, we can write $\theta \in \mathbb{R}^D$ as the direct sum of $Y \in \mathbb{R}^M$, and $Z \in \mathbb{R}^{D-M}$, up to a diffeomorphism $F$, i.e. $\theta = F(Z \oplus Y)$. Hereafter, the diffeomorphism $F$ will be left understood, and we use the equivalent notation $\theta \equiv (Z, Y)$.

slow dynamics only. We further notice that no decimation of the $Z$ variables is needed, as the fast dynamics always remains within a block.

Moreover, we saw that the directions spanned by $Y$ and by $Z$ are locally orthogonal on $\Gamma$, which implies that $\partial/\partial Y$ and $\partial/\partial Z$ can be regarded as the components of the gradient $\nabla_\theta$ orthogonal and tangent $\Gamma$ at any of its points. That is, if $P(Z)$ denotes the projector on $T_{\theta^*}(\Gamma)$, for $\theta^* = (Z, 0)$, then

$$\frac{\partial}{\partial Z} = P(Z) \cdot \nabla_\theta \quad \text{and} \quad \frac{\partial}{\partial Y} = \big(1 - P(Z)\big) \cdot \nabla_\theta \,. \tag{E7}$$

This will be useful later in the derivation.

### E.1.2 PERTURBATIVE SOLUTION

Following the steps in Appendix D.2, we obtain the same hierarchy of equation, equations D7:

$$O(\epsilon^{-1}): \ (\partial_\tau - g \cdot \nabla_\theta) \, h^{(0)} = 0 \tag{E8a}$$

$$O(1): \ \mathcal{M} h^{(1)} = -(\partial_t + \frac{1}{2} \Sigma : \nabla_\theta \nabla_\theta) \, h^{(0)} \tag{E8b}$$

where we can make the dependence of all the terms in the perturbative solution on fast and slow variables explicit:

$$h(\theta, t) \to h^{(\epsilon)}(Z, Y, \tau, t) = h^{(0)}(Y, \tau | Z, t) + \epsilon \, h^{(1)}(Y, Z, t) \tag{E9}$$

As described in Appendix D, it is possible to derive the effective (averaged) dynamics at long timescales from the solvability condition at the next-to-leading order in the multiple timescales expansion.

The leading order, equation E8a, is the deterministic transport equation following the dynamics defined by $\dot\theta = -g(\theta)$. and gives a short-lived transient that occurs over a timescale $O(\epsilon)$, described by the time variable $\tau$, that remains on the manifold $\Theta(Z)$.

Here, we want to capture the dynamics over long times, long after $\theta$ has converged to a point on $\Gamma$. Therefore, we are going to look for the stationary solution of equation E8a:

$$\mathcal{M} h^{(0)} = -g(\theta) \cdot \nabla_\theta h^{(0)} = 0 \tag{E10}$$

assuming that $h^{(0)}$ only depends on the auxiliary variable $t$. Since $g$ is the gradient of the loss $L$, the flow lines described by $\phi$ are locally orthogonal to the contour lines of the loss $L$.

In order to proceed to the next order, equation E8b, we need to impose a solvability condition, as discussed in Appendix D.2. In order to do that, we need to find the left null space of $\mathcal{M}$, i.e.

$$\mathcal{M}^* \, \mu_{\mathrm{inv}}(Y|Z, t) = \nabla \cdot \big(\mu_{\mathrm{inv}}(\cdot|Z, t) \, g\big) = 0 \tag{E11}$$

which is solved by

$$\mu_{\mathrm{inv}}(Y|Z, t) = \delta(Y) \,, \tag{E12}$$

The solvability condition would then read

$$\Big(\partial_t + \frac{1}{2} \Sigma(Z, Y)|_{Y=0} : \nabla\nabla\Big) \psi(Z, t) = 0 \,, \tag{E13}$$

that describes a purely diffusive dynamics.

However, near the manifold, the drift $-g$ can be small enough to compensate the large pre-factor $\epsilon^{-1} = \eta^{-1}$, in such a way that the separation of timescales between the two terms in equation E1 is no longer valid. Then, we would have to expand the deterministic dynamics near $\Gamma$, and consider the small-$Y$ fluctuations which are not captured by the singular leading-order solution, equation E12.

### E.2 EXPANSION IN MULTIPLE TIME AND LENGTH SCALES

In order to deal with the singularity in space generated by the fast convergence to the invariant manifold, in addition to the expansion in multiple timescales, we also need to introduce auxiliary spatial variables, which describe the solution near and far from the invariant manifold $\Gamma$.

As already observed, over timescales $O(1)$ we only expect small fluctuations in the $Y$ variables, $O(\epsilon^\alpha)$, for some $\alpha > 0$. The $Z$ variables, instead, are expected to be always $O(1)$, and do not need a rescaling.

Then, we can replace the variables $Y$, capturing deviations from $\Gamma$, with rescaled variables $y$ that describe small fluctuations:

$$Y \to \epsilon^\alpha\, y\,, \tag{E14}$$

This is analogous to boundary-layer problems, where an inner solution and an outer solution are found near and far from a boundary, to be matched at intermediate scales E (2011).

In these new variables, we get a natural Taylor expansion of the gradient of the loss, $g = \nabla L$:

$$g(\theta) \equiv g(Z, Y) \to g(Z, \epsilon^\alpha y) \simeq \epsilon^\alpha\, H(Z) \cdot y + \epsilon^{2\alpha}\, \frac{1}{2} \nabla H(Z) : yy^\top\,, \tag{E15}$$

where by $H(Z)$ we denote the Hessian of the loss (and $\nabla H(Z)$ its gradient) evaluated at a point $\theta^*$ on $\Gamma$ (i.e. $y = 0$) with coordinates $Z$.

Near $\Gamma$, we expect the fluctuations due to noise (described by the term $\Sigma : \nabla\nabla$ in the generator $\mathcal{L}$, equation C3) are compensated by the elastic term appearing in equation E15. Then, the scaling exponent $\alpha$ is determined in order for this noise-drift compensation to take place at leading order.

The rescaling of the $Y$ coordinates, equation E14, implies

$$\frac{\partial}{\partial Y} = \epsilon^{-\alpha}\, \frac{\partial}{\partial y}\,. \tag{E16}$$

As we have commented on in Appendix B, the leading term in equation E15 is non-vanishing and perpendicular to the tangent space of $\Gamma$ at $\theta^*$ (since all tangent vectors are null vectors of the Hessian $H$, and no other null vector is assumed to exist in the perpendicular direction).

Moreover, as shown in Appendix E.1.1, the derivatives with respect to $Z$ and $y$ are orthogonal on the manifold $\Gamma$, and according to equation E16 we can express

$$\nabla_\theta = \frac{\partial}{\partial Z} + \epsilon^{-\alpha}\, \frac{\partial}{\partial y}\,. \tag{E17}$$

By replacing equation E17 and the Taylor expansion E15 into the expression for the Fokker-Planck generator, equation C3, we notice that the only scaling exponent which allows noise and elastic force to balance in the perpendicular direction is $\alpha = 1/2$. This is indeed the scaling the fluctuations in an Ornstein-Uhlenbeck process with large drift $\epsilon^{-1}$.

With this scaling, and with $\epsilon = \eta$, the generator can be written as

$$\mathcal{L} = \epsilon^{-1}\mathcal{M} + \epsilon^{-1/2}\,\mathcal{L}_{-1} + \mathcal{L}_0 \tag{E18}$$

where

$$
\begin{aligned}
\mathcal{M} &= \Big( -y^j\, H_j^k + D^{ij}\,(1 - P)_j^l\, \frac{\partial}{\partial y^l} \Big)(1 - P)_i^k\, \frac{\partial}{\partial y^k} \\
&= \Big( -y^j\, H_j^k + \frac{1}{2}\, \Sigma_\perp^{kl}\, \frac{\partial}{\partial y^l} \Big)\, \frac{\partial}{\partial y^k}
\end{aligned}
\tag{E19}
$$

$$
\begin{aligned}
\mathcal{L}_{-1} &= \Big( -y^j y^l \nabla H_{jl}^i + \frac{1}{2}\, \Sigma^{ij}\, P_j^l\, \frac{\partial}{\partial Z^l} \Big)(1 - P)_i^k\, \frac{\partial}{\partial y^k} \\
&= \Big( -y^j y^l \nabla H_{jl}^i - \frac{1}{2}\, \big(\Sigma_{\perp,\|} + \Sigma_\|\big)^{ij}\, \frac{\partial P_i^k}{\partial Z^j} + \frac{1}{2}\, \Sigma_{\perp,\|}\, \frac{\partial}{\partial Z^l} \Big)\, \frac{\partial}{\partial y^k}
\end{aligned}
\tag{E20}
$$

$$
\begin{aligned}
\mathcal{L}_0 &= \Big( -y^j y^l \nabla H_{jl}^i P_i^k\, \frac{\partial}{\partial Z^k} + \frac{1}{2}\, \Sigma^{ij}\, P_j^l\, \frac{\partial}{\partial Z^l} \Big)P_i^k\, \frac{\partial}{\partial Z^k} \\
&= \Big( -y^j y^l \nabla H_{jl}^i P_i^k\, \frac{\partial}{\partial Z^k} + \frac{1}{2}\, \big(\Sigma_{\perp,\|} + \Sigma_\|\big)^{ij}\, \frac{\partial P_i^k}{\partial Z^j} + \frac{1}{2}\, \Sigma_\|^{kl}\, \frac{\partial}{\partial Z^l} \Big)\, \frac{\partial}{\partial Z^k}
\end{aligned}
\tag{E21}
$$

where we used the fact $P$ depends on $Z$ only, that $PH = HP = 0$, and where we defined

$$\Sigma_\perp^{kl} = (1 - P)_i^k \, \Sigma^{ij} \, (1 - P)_j^l \tag{E22}$$

$$\Sigma_{\perp,\parallel}^{kl} = (1 - P)_i^k \, \Sigma^{ij} \, P_j^l \tag{E23}$$

$$\Sigma_\parallel^{kl} = P_i^k \, \Sigma^{ij} \, P_j^l \tag{E24}$$

Note that $\mathcal{M}$ is the generator of the Ornstein–Uhlenbeck process in the perpendicular fluctuations $y$.

PERTURBATIVE SOLUTION

In equation E18, the timescale separation is seemingly given by $\epsilon^{1/2}$. In order to be able to match all terms in the generator with the corresponding time derivative, we take

$$t \to \epsilon \, \tau + \epsilon^{1/2} \, \tilde{t} + t \tag{E25}$$

which gives

$$\partial_t \to \epsilon^{-1} \, \partial_\tau + \epsilon^{-1/2} \, \partial_{\tilde{t}} + \partial_t \, . \tag{E26}$$

Similarly, we adapt the perturbative solution, that is

$$h \to h^{(\epsilon)} = h^{(0)} + \epsilon^{1/2} \, h^{(1/2)} + \epsilon \, h^{(1)} \tag{E27}$$

The procedure is then identical to the one described in Appendix D.2. After a short transient, the leading orders give

$$\mathcal{M} h^{(0)} = 0 \, . \tag{E28}$$

It is easy to see that this is solved by a function constant in $y$:

$$h^{(0)}(y|Z, t) = \psi(Z, t) \tag{E29}$$

The corresponding left eigenvector of $\mathcal{M}$, is

$$\mathcal{M}^* \mu_{\text{inv}} = 0 \tag{E30}$$

that is the equation for the equilibrium solution of the Ornstein-Uhlenbeck process:

$$\mu_{\text{inv}}(y|Z) = \frac{\det \Omega^{-1/2}}{(2\pi)^{M/2}} \exp \left\{ -\frac{1}{2} \, \Omega_{ij}^{-1} \, y^i y^j \right\} \tag{E31}$$

where $\Omega(Z)$ is the solution of Gardiner (2009)

$$H(Z) \, \Omega + \Omega \, H(Z) = \Sigma_\perp(Z) \, , \tag{E32}$$

and therefore depends on $Z$.

We can then move to the next-to-leading order, $O(\epsilon^{-1/2})$.

$$\mathcal{M} h^{(1/2)} = -(\partial_{\tilde{t}} + \mathcal{L}_{-1}) \, h^{(0)} \tag{E33}$$

We note that the operator $\mathcal{L}_{-1}$ also has a derivative with respect to $y$ on the right and therefore

$$\mathcal{L}_{-1} h^{(0)} = 0 \, , \tag{E34}$$

due to equation E29. The solvability condition, therefore gives the trivial result $\partial_{\tilde{t}} \psi = 0$, so that we must proceed to the next order. To do that, we need to solve equation E33 for $h^{(1/2)}$. However, we notice that the solvability condition yields an identical equation to E28, which tells us that $h^{(1/2)}$ can be set to 0.

Then, at the last order in the expansion, we have

$$\mathcal{M} h^{(1)} = -(\partial t + \mathcal{L}_0) \, h^{(0)} \tag{E35}$$

The solvability condition for this equation is

$$\left( \partial_t + \bar{\mathcal{L}}_0 \right) \psi(Z, t) = 0 \, , \tag{E36}$$

where the operator $\hat{\mathcal{L}}_0$ is obtained by averaging the coefficients of the generator $\mathcal{L}_0$ over $\mu_{\mathrm{inv}}(y|Z)$. This is indeed very simple: all terms are independent of $y$, except the first which features $y^j y^l$, whose average is $\Omega^{jl}$. This gives

$$\bar{\mathcal{L}}_0 = \Big( -\Omega^{jl} \nabla H^i_{jl} P^k_i \frac{\partial}{\partial Z^k} + \frac{1}{2} \big(\Sigma_{\perp,\parallel} + \Sigma_\parallel\big)^{ij} \frac{\partial P^k_i}{\partial Z^j} + \frac{1}{2} \Sigma^{kl}_\parallel \frac{\partial}{\partial Z^l} \Big) \frac{\partial}{\partial Z^k} \tag{E37}$$

From this equation, we conclude that the average dynamics is given by a drift

$$\bar{f}^k = -\Omega^{jl} \nabla H^i_{jl} P^k_i + \frac{1}{2} \big(\Sigma_{\perp,\parallel} + \Sigma_\parallel\big)^{ij} \frac{\partial P^k_i}{\partial Z^j} \tag{E38}$$

and a noise covariance

$$\bar{\Sigma} = \Sigma_\parallel \tag{E39}$$

that is only parallel to the manifold $\Gamma$.

Note that the second term in the drift E38 involves the derivative along the manifold of the projector $P$, which is solely dependent on the geometry of the manifold.

