# OpenReview forum: "Implicit Regularisation in Overparametrized Networks: A Multiscale Analysis of the Fokker-Planck equation"
_ICLR.cc/2024/Conference — Submitted to ICLR 2024_

### Official Review · Reviewer_57JF · 2023-10-18

**Soundness:** 2 fair
**Presentation:** 1 poor
**Contribution:** 3 good
**Rating:** 3
**Confidence:** 3

**Summary:**

The paper proposes to study the behavior of stochastic gradient algorithms in the viscinity of the invariant manifold by what is referred to as multiscale methods for PDEs. The main contribution is a (somewhat informal) derivation---using the above-mentioned technique---of results previously established by Li, Wang and Arora. To my understanding, this offers a change in perspective by which the authors analyze the evolution of the density of the process instead of the stochastic process directly.

**Strengths:**

- The paper has a clearly delineated goal and is quite crisp.  Indeed, the paper gets to the point very quickly and the main part of the paper is just short of 4 pages. While I think crispness is a clear positive, I think it comes at the cost of clarity here. There is literally no reason to relegate key notation to various appendices when your page count is this low.

- The ideas being introduced in this paper are likely of interest to the community. I very much like the  idea of analyzing the density and associated PDEs via perturbation theory instead of going via the SDE (where the SDE to some extent is a more technically challenging object). I think simplification is a very worthy goal.

**Weaknesses:**

By my estimation, this paper's contribution is the introduction of a new proof technique to study gradient algorithms. I am quite sympathetic to what the authors are trying to do here but think the paper ultimately falls short of this goal and I cannot recommend acceptance in it its current form. My reasoning here is based on that I believe the paper is mainly to be seen as educational/expositional. Indeed, the results in their paper are not novel as the authors themselves state. I do not view this as an issue in itself but it does mean that the paper's contribution is solely in terms of exposition of an analysis approach. For this reason, I think the paper should to some extent be seen as educational and therefore (in place of novelty) has to meet a higher standard in terms of expositional clarity than a conference paper othewise would. I outline below why I think it falls short of this standard.

* While the voice and overall language is good, the manuscript is written in quite poor mathematical style. My main grievance is with  section 2 "the main result", wherein we find a sequence of unjustified approximations leading to an expression analogous to that derived by Li, Wang and Arora. Instead of providing actual justifications of the steps taken an appendix of various  related facts is handed to us from which we are---in Appendix E---to understand that the claimed result (11) is correct. It is quite unclear at first reading how various implications are justified by those preceding them. This is unfortunate since I think the ideas in this manuscript are quite interesting and eventually worthy of publication, but this style is doing your readers a huge disservice. I think the paper needs to rewritten in a way such that readers can follow how one claim follows logically from the preceeding one and so forth. A few examples:
     * The paragraph beginning with "We can introduce auxiliary variables...". Instead of stating that you can, just do it and show how it is done or at least reference someplace doing so. What are the dynamics of y? This might be individual preference but I would prefer to see them over just the generator. It may also be helpful to show---in math---how they relate to the original process.
     *In appendix E you state that "[a]s we have commented on in Appendix B" to justify a certain local orthogonality relation. First, it would be helpful to point to exactly where (\eqref{}?). Second, I think "commenting" on something is insufficient to establish a claim. Just state exactly why one implication follows from the next. I see no need for this ambiguity.
     * While on the topic of references, it is quite unhelpful to simply reference a book of 350 pages or so. Could you please reference exactly which theorem or statement you are using?
     * In the appendix the authors list quite a few claims and it is not clear to me that all of them are relevant to the article. Given that the authors have chosen to write a very short article, I encourage them to be consistent in this goal and not list facts that are not pertinent to the main development. This actually somewhat compounds the issue of the main derivation being hard to follow---it was unclear to me at first reading which parts of the appendix are needed for this and which are not.


* A few other remarks on style: There is quite a bit of notation in the intersection of undefined and nonstandard (at least for an ML theorist), e.g. : for trace inner product, that  isomorphism symbol for approximation, perp and parallell and so and so forth. If you are not going to define your notation in the beginning of the article, _at least_ collect it somewhere in one place.  Finally, stating  assumptions at the very end of a derivation of a claimed result (non-degeneracy of $H$ on $\Gamma$) hinders the overall logical flow.

**Questions:**

Please see weaknesses above for a few more clarifying questions.

* Following (A8) what is the justification for this approximation? Why are we allowed to pass from a possibly highly dependent object to the most benign situation of orthogonal Gaussians?

* What does the notation in (E14-E15) mean? Is $\rightarrow$ implication?

*  Is it required that the manifold $\Gamma$ has constant rank? Can one imagine situations of interest in which this is not the case?

---

### Official Review · Reviewer_a3pb · 2023-11-01

**Soundness:** 2 fair
**Presentation:** 1 poor
**Contribution:** 2 fair
**Rating:** 3
**Confidence:** 3

**Summary:**

The authors study the optimization behavior of stochastic gradient descent for over-parameterized networks based on a multiscale analysis for elliptic PDEs. Specifically, they provide an alternative way of deriving the same result by [Li et al. (2022)]. This result clarifies the two-phase dynamics of SGD: in the first phase, SGD moves to the manifold of minimizers, and in the second phase, SGD moves along this manifold due to the drift caused by stochastic gradient noise.

**Strengths:**

The alternative way of showing the two-phase dynamics of SGD [Li et al. (2022)] seems technically interesting and could be potentially useful for future studies.

**Weaknesses:**

- While I acknowledge the potential utility of this work, my primary concern relates to the quality of the manuscript. Given that there are approximately 5 remaining pages for the main body, there is room for improvement. Specifically, more detailed explanations of the theory would enhance the paper's accessibility.

- The advantage of using the multiscale technique in comparison to Li et al. (2022) remains somewhat unclear. What motivates the introduction of a new technique in light of existing work? Although providing new proof is technically important, an additional comment regarding the advantage is necessary to persuade the reader of its significance.

**Questions:**

Is there any new implication suggested by this work?

---

### Official Review · Reviewer_NSHD · 2023-11-04

**Soundness:** 2 fair
**Presentation:** 2 fair
**Contribution:** 2 fair
**Rating:** 3
**Confidence:** 3

**Summary:**

The paper addresses the dynamics of over-parametrized networks during the training process, where it's observed that a manifold of solutions with minimal empirical loss exists. The authors extend the work of Li et al. (2022) by providing a new derivation of the learning dynamics as a stochastic differential equation (SDE) in the limit of a small learning rate. They describe a two-phase learning process: an initial deterministic phase driving the parameters toward the manifold, followed by a phase where noise introduces a deterministic drift along the manifold, which acts as an implicit regularizer by favoring smoother solutions. This drift is conceptualized as an averaging of the dynamics over the noise perpendicular to the manifold, framed as an Ornstein-Uhlenbeck process. The paper builds upon intuitive arguments previously suggested by Blanc et al. (2020) and uses an averaging of the Fokker-Planck equation to describe the dynamics over this quasi-stationary state. This result demonstrates the application of multiscale methods for elliptic partial differential equations (PDEs) (Pavliotis and Stuart (2008)) to optimization problems in machine learning.

**Strengths:**

1. This paper studies the implicit regularization of Stochastic Gradient methods in overparametrized models, which is an important topic and very relevant to ICLR. It presents a multiscale analysis for the result in Li et al., 2022. The analysis follows the intuition of Blanc et al., 2020, but is more general than Blanc et al., 2020 in the sense that it can deal with general noise covariance.

2. Though I am not very familiar with the literature of multiscale analysis, the derivation makes sense to me. Though no new results are presented in this work, this method presented in the paper could probably be applied to other multiscale problems in deep learning, e.g., those related to grokking (Power et al.,2022).

**Weaknesses:**

1. The derivation in this paper is nor rigorous. It is not clear what assumptions are needed for the loss function and the gradient noise covariance.

2. Singular perturbation is not well-defined throughout the paper, though from Appendix E it seems to be an important issue.

3. The authors claim the result is consistent with Li et al., 2022, but it is not very clear how to recover Eq. (16) from Eq (14a,14b)

4. Some non-standard notations are used without definition, e.g., $P\cdot \nabla P$ in Eq (13).

Typos:

(1). Eq(14b): $\overline f^i$ should be $\overline f^k$. $\partial \theta^*$ should be $\partial \hat \theta$

(2). In Eq(20), second row, there are unmatched subscripts (superscripts) $l,k$.

**Questions:**

See point 2 and 3 in the weakness.

---

### Official Review · Reviewer_3Hfp · 2023-11-05

**Soundness:** 3 good
**Presentation:** 2 fair
**Contribution:** 2 fair
**Rating:** 5
**Confidence:** 4

**Summary:**

This paper provides an alternate derivation of the results in Li et al. (2022), which characterize the implicit regularization of SGD with small learning. The presented argument demonstrates the application of multiscale methods for elliptic partial differential equations to optimization problems in machine learning.

**Strengths:**

The implicit regularization of optimization algorithms is an important topic, and this paper presents a new technical tool through the lens of multiscale methods. The heuristics behind the argument are clear and yield an alternate approach by analyzing the generator of the SDE approximation of SGD, different from the original proof based of distribution of the SGD trajectory in Li et al. (2022).

**Weaknesses:**

1. The main text seems a bit too technical, and might be a bit hard for the broad audience to digest. I would suggest the authors to revise it a bit to help the reader understand the terminologies.
2. The argument in the paper can be applied to the SDE approximation, i.e., equation (2), but not the original discrete SGD.
3. The argument in equation (5)~(10) seems to be only a local analysis.
4. There are no end-to-end results. What kind of conclusions on SGD can be deduced from the derived average generator? It would be helpful if the authors can elaborate on the more broad implications of the current argument, as well as other applications.

**Questions:**

1. In the paragraph after equation (5), is $\Phi(\theta_0)$ the same map induced by gradient flow as that in Li et al. (2022)?
2. Is it possible to directly apply this kind of stochastic homogenization technique to the discrete updates?

---

### Meta-Review · Area_Chair_QbuR · 2023-11-29

**Metareview:**

The paper considers implicit regularization of SGD of over-parametrized networks, based on multiscale analysis of the Fokker-Planck equation for the training dynamics. It provides an alternative approach to implicit regularization based on PDE techniques. While the reviewers find the result interesting, they also point out some shortcomings of the result and the presentation. The authors fail to respond to reviewers' comments during the discussion stage.

**Justification For Why Not Higher Score:**

Authors fail to engage in discussions with the reviewers or provide a revision addressing the weakness.

**Justification For Why Not Lower Score:**

N/A

---

### Decision · Program_Chairs · 2024-01-16

Reject